# Potential Roles and Functions of Listerial Virulence Factors during Brain Entry

**DOI:** 10.3390/toxins12050297

**Published:** 2020-05-05

**Authors:** Franjo Banović, Horst Schroten, Christian Schwerk

**Affiliations:** Department of Pediatrics, Pediatric Infectious Diseases, Medical Faculty Mannheim, Heidelberg University, 68167 Mannheim, Germany; franjo.banovic@medma.uni-heidelberg.de (F.B.); horst.schroten@umm.de (H.S.)

**Keywords:** *Listeria monocytogenes*, virulence factors, internalin, autolysin, listeriolysin, brain invasion

## Abstract

Although it rarely induces disease in humans, *Listeria monocytogenes* (*Lm*) is important due to the frequency of serious pathological conditions—such as sepsis and meningitis—it causes in those few people that do get infected. Virulence factors (VF) of *Lm*—especially those involved in the passage through multiple cellular barriers of the body, including internalin (Inl) family members and listeriolysin O (LLO)—have been investigated both in vitro and in vivo, but the majority of work was focused on the mechanisms utilized during penetration of the gut and fetoplacental barriers. The role of listerial VF during entry into other organs remain as only partially solved puzzles. Here, we review the current knowledge on the entry of *Lm* into one of its more significant destinations, the brain, with a specific focus on the role of various VF in cellular adhesion and invasion.

## 1. Introduction

*Listeria monocytogenes* (*Lm*) is a gram-positive opportunistic pathogen of humans and animals with a worldwide distribution, known for its ability to invade human cells (both phagocytes and non-phagocytes), cross multiple body barriers and cause listeriosis, a potentially lethal disease [1]. *Lm* is a genetically diverse species divided into several lineages, serotypes and clonal complexes (CC). There are at least four currently recognized lineages, although most of the clinically relevant strains belong to lineages I and II [2]. Three strains most commonly used in research—EGD, EGDe and 10403S—all belong to serovar 1/2a of lineage II [2]. CC classification utilizes multilocus sequence typing (MLST), where bacterial isolates are assorted into distinct groups based on the similarity of sequences of selected housekeeping genes [2,3]. Through use of MLST, *Lm* was divided into several more prominent and many less prominent CC, all of which could be correlated with specific serovars (e.g., serovar 1/2a and CC7) [2,3,4]. Interestingly, it has been observed that *Lm* strains and isolates commonly found in food belong to CC9 and CC121, while most strains and isolates most often obtained from infected people belong to CC1, CC2, CC4 and CC6, outlining a division between food associated and clinically associated strains [2].

The main entry gate for listerial infection is the gastrointestinal tract, since the main vector for *Lm* is contaminated food. In immunocompetent people, efficient cell-mediated immune response leads either to an asymptomatic infection or a self-limited gastroenteritis, resolving within days [1,5,6,7]. In immunocompromised people—including pregnant women, newborn children, elderly people and organ transplant recipients—the infection is often much more severe [1,7]. The bacteria invade the cells of the gut epithelium, cross the gut barrier and enter the bloodstream, both independently and by invading the circulating phagocytes (mainly macrophages) [1,8]. They are then carried by blood to the liver, where resident macrophages eliminate most of them with a minority surviving within hepatocytes; and to the spleen, where they are present mainly within phagocytes [1]. Although the infection can be eventually cleared by the immune system in some cases, usually it will linger unless treated with antibiotics. The longer the infection persists, the greater are the chances that *Lm* will spread further within the body—invasion of the central nervous system (CNS), specifically the brain, as well as invasion of the fetus in pregnant women are among the most dangerous outcomes of such prolonged infection [1,7]. 

*Lm* has been known as a CNS-invading organism for decades, but the primary path (or paths) it undertakes to enter the brain are still not definitely confirmed [6,7,9]. The main obstacle for the clarification of this issue is the difficulty of detecting the entry of bacteria into the brain—when the signs of the CNS infection become apparent, *Lm* is already present in multiple parts of the brain. Post-mortem analysis of infected individuals (or animals) is also limited when it comes to identification of the entry spots, for the same reasons. Investigations done so far indicate that *Lm* could infiltrate the brain from either the blood or the nerves connected to peripheral tissues [7,10,11,12,13,14,15] (Figure 1).

An established bacteremia is a prerequisite for blood-borne listerial invasion of the brain, which almost universally manifests as either meningitis or meningoencephalitis [7,10]. There are two possible hematogenous routes that can be utilized: via the blood-brain barrier (BBB) or via the blood-cerebrospinal fluid barrier (BCSFB) [9,16,17]. Penetration of the blood-brain barriers could be achieved either directly or via a “Trojan horse” mechanism, where *Lm* is carried across the barrier by infected phagocytes [18]. There are several publications showing that *Lm* can successfully invade endothelial cells of the brain microvasculature (which—together with astrocytes and pericytes—comprise the BBB) and epithelial cells of the choroid plexus (which form the BCSFB) in vitro, as well as transmigrate through monolayers of these cells [11,17,19,20,21]. In vivo studies showing the same are much scarcer, but include examples of “Trojan horse” spread of bacteria through infected monocytes [8,21].

*Lm* can also invade the brain by retrograde axonal transport [7,9,16]. Two different routes using this mode of intrusion into the brain have been described so far—one that utilizes the cranial nerves for the entry (primarily the trigeminal nerve), and the other which exploits the olfactory epithelium [7,9,14,15,16]. Unlike the hematogenous routes of entry into the brain where the path undertaken by bacteria from the gut is the same until the very end, these two are quite different. The transport using cranial nerves normally occurs in immunocompetent people—an exception to the norm of listerial CNS invasion being connected to compromised immune response—when the bacteria enter through wounds in the oral mucosa, infect the local macrophages and/or recruited phagocytes and spread to the cranial nerves; it manifests as rhombencephalitis [9,14]. Experimental proof for this scenario was provided by Antal and colleagues, who showed that mice developed rhombencephalitis after inoculation of either facial muscles or endings of facial nerves with *Lm* [22]. Additionally, the side of the brain stem which was first affected correlated with the side of the face where bacteria were injected [22]. On the other hand, invasion through the olfactory epithelium occurs during birth, when the newborn comes into contact with the bacterial flora of the mother’s vaginal tract; it manifests as meningitis [15]. The majority of research concerning the cranial nerve route was done in ruminants and it showed involvement of trigeminal nerves in invasion of the CNS [14,23]. A single study analyzing the olfactory epithelium route was done in newborn mice, and showed much higher propensity for CNS infection when *Lm* was applied nasally than when it was applied orally [15].

It is a well-documented fact that different manifestations of neurolisteriosis vary in incidence between humans and domestic animals—primarily sheep and goats, but also bovines—and it is also known that neurolisteriosis in general is much more common in these animals than in humans [16,23]. In humans, meningitis/meningoencephalitis is the typical form of listerial CNS infection, with brain abscesses and rhombencephalitis being less common; on the other hand, rhombencephalitis is observed in domestic animals much more often than other forms of neurolisteriosis [16,23]. Since it is currently believed that meningitis is the outcome of a blood-borne listerial invasion of the brain while rhombencephalitis occurs after the incursion of bacteria that have invaded neural periphery in the oral cavity, it could be hypothesized that the hematogenous route is preferred in humans and the neurogenous route in ruminants and cattle.

## 2. Virulence Factors

As an opportunistic pathogen capable of parasitizing on multiple different cell types and passing across several cellular layers of the body, it is unsurprising that *Lm* is equipped with a plethora of virulence factors (VF) enabling its lifestyle [1,24]. Not all of them are involved in cellular invasion, however, and not all invasion-related VF have been connected to listerial entry into the brain. Brain entry-related VF can be divided into two groups: members of internalin family (Inl) and other VF (Figure 2).

### 2.1. Internalins

Internalins are a protein family specific to genus *Listeria*. There are 25+ currently known Inl family members in *Lm* strain EGDe, which is the number most commonly quoted in reference to Inl family size [1,24,25,26]. Other Inl or Inl-like proteins have been found in other strains and clinical isolates, however, and the number of newly described Inl family members continues to grow [26,27]. Two features common to the entire Inl family are a Sec-dependent N-terminal signal peptide and a leucine-rich repeat (LRR) domain [26]. Presence of a signaling peptide points to extracellular localization—internalins can be bound to the bacterial surface by a covalent bond with peptidoglycan (majority of Inl proteins, e.g., InlA), by a loose connection to a cell wall component such as lipoteichoic acid (e.g., InlB), or not be bound at all (e.g., InlC) [26]. LRRs are binding motifs more often found in eukaryotes than prokaryotes that usually participate in protein-protein interactions—they can be found in multiple copies in each Inl (from 3 repeats in Lmo2445 to 28 in InlI), suggesting that internalins are able to bind to the proteins of the host [26,28].

Although the majority of Inl proteins are relatively poorly described and characterized, several members of the family were investigated to at least some extent, with InlA and InlB being the ones most thoroughly studied. Genes coding for these two proteins are part of a same operon on the bacterial chromosome, whose transcript can contain either InlA or both InlA and InlB (separated after translation) [29]. Both of them were found to be required for invasion into multiple different cell types, initiating a zipper-type endocytosis [11,30,31]. In addition to InlA and InlB, there are other Inl family members addressed in the literature—for some of them, host cell interaction partners and functions have been described (InlC, InlF, InlK and InlP), while the role of the others is suggested on the basis of available data but not yet fully confirmed (InlH, InlJ and InlL) [19,21,26,32,33,34,35].

Similarity of the basic structure between the members of the Inl family suggests that all of them might be VF/modulators, possibly with a role in cellular adhesion and/or invasion [26]. There is evidence that *Lm* mutants with deletions or combinations of deletions of specific internalins invade non-phagocytic cell lines (such as Caco-2 and HBMEC) less effectively as well as show defects in infection experiments in animal models—curiously, some deletions even seemed to enhance the invasion [19]. Taken together, it is very likely that the interactions between various members of the Inl family and their dependency on each other might be much more complex and important than what is apparent at first glance.

#### 2.1.1. Internalin (InlA)

The first described Inl protein, InlA, is a membrane-bound protein that binds to the adherens junction protein E-cadherin (Ecad) found in several epithelial tissues [1,7,29]. Unlike its Inl family relative, InlB, InlA has a role not only in invasion of the host cells but also in adhesion [36]. InlA was shown to be essential for the penetration of the gut barrier—a deletion mutant is strongly attenuated in an oral (but not intravenous) mouse infection model, and clinical isolates of *Lm* not possessing a full length, functional InlA are rare [29,30,37,38,39]. 

The role of InlA during brain invasion is less well defined. A location where it could play a more important role is the epithelium of the choroid plexus [17,20,40,41]. Indeed, deletion of InlA causes a decreased invasion into human epithelial choroid plexus papilloma (HIBCPP) cells in vitro [17]. The deletion of InlB has a similarly strong effect, however, but the deletion of both InlA and InlB does not decrease the invasion further, implying that—unlike what is observed in the gut epithelium—both InlA and InlB are required for entry into the choroid plexus epithelium [17]. Interestingly, there are indications that InlA-mediated invasion is dependent on support of several other members of the Inl family other than InlB—InlC and members of InlGHE cluster (InlG, InlH and InlE)—for invasion into gut epithelium, but there is no data available concerning the invasion into the choroid plexus epithelium [19].

Binding of InlA to Ecad initiates downstream signaling leading to actin polymerization and zipper-type endocytosis of the bacteria [42]. This process can be divided into four parts: 1. posttranslational modification of Ecad; 2. recruitment of proteins required for clathrin-mediated endocytosis; 3. first wave of actin polymerization; 4. second wave of actin polymerization [42,43]. Posttranslational modification of Ecad begins after the binding of InlA and consists of phosphorylation (through kinase Src) and ubiquitination (through ubiquitin ligase Hakai) [42,43,44,45,46]. Phosphorylated and ubiquitinated Ecad then participates in the recruitment of proteins required for clathrin-mediated endocytosis (such as dynamin and clathrin itself) [42,47,48]. Following this is the first wave of actin polymerization through dynamin, Src and cortactin, resulting in the recruitment of Arp2/3 complex, the main player of the actin reorganization machinery [42,46,48]. Finally, the second wave of actin polymerization takes place, reliant on interaction between α and β catenins with Arp2/3 complex [42,48,49].

An interesting detail to note in relation to InlA-Ecad interaction is its species specificity due to subtle differences in Ecad build between species: while InlA can bind well to human or guinea pig Ecad, interaction with mouse or rat Ecad is much weaker [50]. This is important for considerations of potential risk of listeriosis in non-human hosts (especially domestic animals), as well as during generation of in vitro and in vivo models.

#### 2.1.2. Internalin B (InlB)

InlB has a unique characteristic within the Inl family; since its GW motifs (with a conserved Gly–Trp (GW) dipeptide) provide it with only a loose bond with the bacterial surface, it can be found both bound and in free form [26,29,42,51]. It has multiple interaction partners: receptor tyrosine kinase Met, also known as hepatocyte growth factor (HGF) receptor; universally expressed protein gC1qR, connected to the complement cascade; and cell surface glycosaminoglycans [42,52,53,54]. All of them have been connected to cellular invasion, but a direct, active role has so far been decisively proven only for Met, while the role of the others is not yet clarified [42,53,55,56]. Met has a much wider tissue distribution than Ecad, which is limited solely to a number of epithelial tissues—consequently, InlB was found to be important for entry into more different cell lines (such as hepatocytes, fibroblasts and endothelial cells) than InlA [11,30,51,55,56,57]. Interestingly, InlB (along with two other internalins, InlC and InlJ) binds to MUC2, a secreted mucin that can be found in intestinal mucus, but not to MUC1, a cell-bound mucin [35]. Further investigation of interactions of various internalins with mucins—which are found not only in the epithelium (and the mucus lining it) of the digestive tract but also in other organs—would surely be of interest.

Concerning the importance for brain invasion, interaction of InlB and Met could play a role in crossing of either BBB and/or BCSFB, as it was shown in in vitro studies that *Lm* is able to infiltrate cell lines representing both blood-brain barriers [11,17,20,58]. As described earlier, invasion of *Lm* into HIBCPP cells (expressing both Ecad and Met) is dependent on both InlA and InlB [17]. However, when cell lines expressing only Met are used—such as the brain microvascular endothelium cell line HBMEC—deletion of InlB alone causes a stronger drop in invasion than either the deletion of InlA, InlB or both in HIBCPP cells, indicating that InlB is the main entry-related VF in HBMEC [11]. A connection between this dependence of *Lm* on InlB for brain invasion and occurrence of listerial meningitis exclusively in immunocompromised people was made in a study which reported a strong impairment of listerial invasion in HBMEC cells in presence of adult human serum (containing anti-*Lm* antibodies) which was not observed if fetal human serum was used [59]. Curiously, deletion of any of the genes from the InlGHE cluster significantly increased the InlB-dependent invasion into HBMEC cells. Experimental results showed that this increase in invasiveness can be tracked to enhanced transcription of InlB, but only in the case of the deletion of the whole InlGHE cluster: there was no increase in transcription or protein presence of InlB in single and double mutants [19].

The downstream signaling process of binding of InlB to Met follows the same four steps described for the binding of InlA to Ecad, although with some differences [42]. Upon contact with Met, InlB dimerizes and thus prompts the dimerization of Met as well, causing its autophosphorylation, and phosphorylated Met is further ubiquitinated by ubiquitin ligase Cbl [42,60,61]. Recruitment of the clathrin-dependent endocytosis machinery functions in the same way as in InlA-induced signaling, but the two actin polymerization wave steps are different: the first wave is initiated by dynamin and cortactin (without involvement of Src), while the second wave utilizes a more complex interplay of PI3K, several Rho GTP-ases (such as Rac1 and Cdc42) and actin-binding proteins (different for different cell lines) for recruitment of the Arp2/3 complex [42,60,62,63,64,65,66,67]. An important part of Met-conveyed signal transduction is the interaction with its co-receptor, CD44v6 [68]. Experiments done in HeLa cells imply that Met signaling caused by binding of HGF cannot be initiated without the contact of HGF to both Met and CD44v6 [69]. CD44v6 might also be involved in Met signaling caused by InlB, which functionally mimics HGF [56,68,70]. The latter findings are controversial, however, since results obtained by Dortet and colleagues did not point to a connection of CD44 (or CD44v6) to Met-related signal transduction [71]. Still, there is no real confirmation of the importance of this interaction in an in vivo setting. 

As for InlA, the affinity of InlB for its binding partners also seems to be species specific—InlB connects well with human or mouse Met and gC1qR, but interaction with guinea pig or rabbit Met and gC1qR is severely hampered [72]. Consequently, the same considerations concerning epidemiology and experimental model usage as those mentioned for InlA should be taken into account.

#### 2.1.3. Internalin F (InlF)

Although discovered only a few years after InlA and InlB, InlF did not garner as much attention as the more distinguished members of the Inl family—while the deletion of either InlA, InlB or both caused a pronounced decrease of invasion into the majority of cell lines used, the deletion of InlF did not seem to affect cellular invasion levels at all [73]. This observation was somewhat puzzling considering the relative similarity in structure between InlA and InlF [73]. It was much later that it was shown that a certain condition—inhibition of Rho-associated protein kinases (ROCKs), which have multiple cellular roles including actin cytoskeleton regulation [74]—enhances the adhesion to and invasion of *Lm* into several human and murine cell lines, and that it also increases the bacterial loads of *Lm* in liver and spleen of infected mice [75]. Curiously, the same work indicated that the deletion of InlF prevents this ROCK inhibition-mediated increase of invasion in mouse (but not human) cell lines, as well as the increase of listerial load in organs of mice [75,76].

Recently, cell surface-associated vimentin—an intermediate filament found in a number of tissues of mesenchymal origin—was proposed as the interaction partner of InlF [21]. This interaction was investigated both in vitro and in vivo and implicated as immensely important for listerial entry into the brain under the conditions of ROCK inhibition, as well as unrelated to InlA- and InlB-mediated pathways. Deletion of InlF caused a decreased invasion in the human brain microvascular endothelial cell line hCMEC/D3 in comparison to both wild type bacteria and InlA and InlB mutants, as well as a decreased load in the brain (but not liver and spleen) of the infected mice [21]. The underlying mechanism of this interaction is still unknown. Another study done in HMEC-1 cells showed no dependence of *Lm* on InlF during cell entry, however, proposing the stiffness of the surface (or extracellular matrix, in in vivo conditions) as the main factor of entry [77]. The results of the two groups are not necessarily mutually exclusive, however, since there can be considerable difference between various vascular endothelial cell lines, which is reflected in the preferred paths that *Lm* or other intracellular pathogens will utilize to enter them. 

All the information assembled so far suggests that InlF could be a backup system for bacterial invasion into the brain activated only under specific conditions [21,73,75]. Activity of ROCKs has been associated with several illnesses, such as neurodegenerative and cardiovascular diseases, diabetes and cancer, and multiple ROCK inhibitors are being looked into (or have been already approved) as potential treatment [78]. Consequently, it becomes obvious that InlF-mediated *Lm* hyper-invasiveness could be a potentially serious risk factor that has to be taken into account in any medical treatment which utilizes ROCK inhibitors.

#### 2.1.4. Internalin J (InlJ)

As mentioned earlier, all Inl family proteins contain an LRR domain with a various number of LRR motifs. One of the peculiarities of InlJ is the fact that its LRRs contain cysteine, making them different from those of almost all other Inl proteins. [79,80]. It is also one of seven Inl family members known to possess a mucin-binding protein (MucBP) domain, which might play a role in binding to host mucins [26]. Some reservations were expressed about the role of the MucBP domain in these interactions, however, since it was demonstrated in a study that both the internalin domain of InlJ alone and other internalins lacking a MucBP domain were able to bind MUC2 [35]. Another detail that makes InlJ stand out is its selective expression by *Lm*. The majority of listerial VF are readily expressed under standard in vitro growth conditions, but InlJ is one of the exceptions [81]. Although its mRNA can be found in bacteria grown under several different conditions—in medium, in in vitro cultures or in mice—the actual protein presence was observed only in bacteria recovered from infected animals [81]. Consequently, the deletion of InlJ has no effect on invasion rates into cell cultures, but it causes a decrease in virulence in both intravenously and orally infected mice [81].

Published work aimed specifically at the connection between the brain invasion and InlJ is scarce. There is a publication, however, that reports a high correlation of the presence of functional InlJ and the occurrence of rhombencephalitis in ruminants, suggesting a role in the development of this disease [82]. 

Concerning interaction partners of InlJ, the previously mentioned study by Lindén and colleagues showed that purified InlJ binds to MUC2 and does not bind to MUC1 [35]. Additionally, in vitro experiments with *Listeria innocua* (*Li*)—the non-pathogenic close relative of *Lm* lacking two pathogenicity clusters found in *Lm*, *Listeria* pathogenicity island 1 (LIPI-1) and InlAB cluster [83]—demonstrated that *Li* transfected with a vector carrying InlJ was able to adhere to placental JEG-3 cells but not to invade them, unlike *Li* expressing InlA, which was able to do both [81]. Taken together, these findings suggest that InlJ might play a role in bacterial adhesion.

#### 2.1.5. Internalin L (InlL)

In a study by Autret and colleagues, signature-tagged mutagenesis (STM) was used to generate transposon mutants of *Lm* EGD which were screened in mice for changes in virulence, with a specific focus on brain infection—one of the mutants which were selected for more detailed investigation was a ORF626 mutant [84]. ORF626, later known as Lmo2026 and currently as InlL, was—based on the sequence of the gene encoding it—predicted to have an Inl-like structure, which was later confirmed [26,84,85]. It is a bacterial surface-bound internalin, and, similarly to InlJ, possesses a MucBP domain [26]. In in vitro experiments, the ability of an InlL deletion mutant to invade and replicate within cells was tested in multiple cell lines—epithelial cells (Caco-2, HeLa and Vero), hepatocytes (HepG-2) and macrophages (J774). Invasion and replication rates were either the same as the wild type or somewhat decreased, except for the HepG-2 line, in which they were significantly decreased [84]. Curiously, InlL seems to be absent not only from all non-pathogenic *Listeria* species, but also from most known strains of *Lm* [26]. 

There is no data on the potential role of InlL in brain invasion except for that presented in the study which originally discovered it. Mice were infected with wild type *Lm* and several deletion mutants including an InlL mutant—while the mice infected with wild type bacteria died before the end of the experiment, the mice infected with deletion mutants recovered by the end of the experiment [84]. All of the tested deletion mutants were quickly cleared from the brain, but the InlL mutant was the only one which persisted in both the liver and the spleen until the end of the experiment, indicating that InlL was not necessary for the entry into these two organs and pointing at a possible specific role it might play in the breaching of the brain [84].

Presence of a MucBP domain in InlL brought about the question of possible interactions of InlL with mucins, which also implies a role in bacterial adhesion [26]. It was demonstrated by Popowska and colleagues that InlL binds MUC2 but does not bind MUC1—something already observed for InlB, InlC and InlJ—but the study did not further investigate the significance of the MucBP domain for this interaction, leaving the issue unresolved [26,84,85].

### 2.2. Other Virulence Factors

#### 2.2.1. Auto

Member of a group of bacterial hydrolases known as autolysins, Auto is a surface protein which contains GW motifs that mediate its association with the bacterial surface, similar to InlB [24,86,87]. The human body is a dynamic and frequently changing environment that requires a well-coordinated exchange of key surface proteins on the part of *Lm* to be navigated successfully. Most of the surface proteins are bound to bacterial peptidoglycan, which makes proteins that contribute to its remodeling vital for efficient survival and spread of *Lm* within the host. Auto contains an N-terminal N-acetylglucosaminidase domain which is activated by proteolysis at low pH, highlighting its role in peptidoglycan rearrangement [24,88]. It was the only autolysin found in *Lm* which was absent from the genome of *Li*, and its deletion mutant showed a strong decrease of invasion (but not adhesion to cells) both in vitro (Caco-2, Hep-2 and Vero cell lines used for both adhesion and invasion tests, and GPC16 and L2 cell lines used only for invasion tests) and in vivo (oral infection of guinea pigs and intravenous infection of mice) [24,86,88].

There is no clear evidence for involvement of Auto in brain invasion, but a study in mice by Cabanes and colleagues reported lower bacterial loads for Auto deletion mutant in all observed organs, including the brain [86].

Its activity as a modifier of the bacterial wall, taken together with its exclusive presence in *Lm*, suggests that Auto has a supportive role during host barrier penetration—regulating the presence of different surface VFs—rather than an active one [24].

#### 2.2.2. IspC

Another bacterial surface autolysin, IspC shares the mode of cell wall association through GW motifs with Auto, although it contains seven module repeats rather than the three possessed by Auto [89,90]. A deletion mutant of IspC showed several differences in comparison to wild type bacteria both in vitro and in vivo. In in vitro experiments, it had a much lower surface presence of several VFs—including InlA and InlB—although their expression in the cell did not change [90]. Curiously, it also showed a cell-type dependent deficiency in both adhesion to and invasion into multiple tested cell lines [90]. In animal experiments, the mutant was not lethal for the tested mice for the duration of the experiment while the wild type *Lm* was, and it accumulated in a significantly lower number in all observed organs except the spleen [90].

When it comes to the CNS invasion, IspC is one of the rare listerial VFs which have been connected to the penetration of the BCSFB but not the BBB. Whereas the IspC deletion mutant was both adhering to and invading HBMEC at the same rate as the wild type *Lm*, both its adhesion and invasion rates were decreased by 50% in the sheep choroid plexus cell line SHP [90]. The mutant also reached a significantly lower bacterial load in the brains of infected mice in comparison to the wild type bacteria [90].

Although multiple listerial autolysins have been implicated in pathogenicity of *Lm*, only a few of them were directly connected to any specific role in it. IspC seems not to only play an indirect role as a regulator of presence of other VFs—such as what is currently known of Auto—but also acts as an adhesin in its own right, which was further confirmed by experiments in which its purified form was binding to the same cell lines to which *Lm* with IspC deletion was unable to adhere [90].

#### 2.2.3. Listeriolysin O (LLO)

LLO is one of the best known listerial VF, and one of the few actual toxins produced by *Lm* [24,91,92]. It is mostly a secreted protein, although it can be also found bound to the bacterial surface. The primary mode of action of LLO is the disruption of various cellular membranes through insertion followed by oligomerization and formation of pores. Cholesterol was identified as its plasma membrane receptor, placing LLO in the family of cholesterol-dependent cytolysins (CDC) [91,92]. Most CDCs primarily act from outside of the cell, creating holes in the plasma membrane that disrupt it and lead to cell death [93]. LLO is different, however; although it is also active before the bacterial internalization into the cells, its optimal activity is in acidic surroundings (pH = 5.5) of the phagosome, contributing to the escape of *Lm* into the cytoplasm [91,92,94]. Decreased activity in more basic surroundings of the cytosol as well as the presence of a PEST (“rich in proline (P), glutamic acid (E), serine (S) and threonine (T)”) sequence at the N-terminus of LLO—which interacts with Ap2a2 of the AP2 complex and thus promotes endocytosis of LLO—prevent it from causing damage to the plasma membrane from the inside, allowing *Lm* to continue with its intracellular parasitism [91,92,95]. As expected, the deletion of *hly*, the gene encoding for LLO, strongly attenuates listerial virulence both in vitro and in vivo, making the deletion mutant basically avirulent [91,92,96,97]. The list of processes linked to the activity of LLO does not end there, though: it was also presented as an inducer of downstream signaling (e.g., MAPK and NFκB pathways), modifier of morphology of mitochondria and ER, and a participant in the internalization of *Lm* into various cell lines [91,92,98,99,100,101].

In relation to the invasion of the brain, there are two proposed roles that LLO might play: 1. penetration of the blood-brain barriers during listerial bacteremia; 2. penetration of the olfactory epithelium and spread to the brain in newborns infected via the nasal route during birth [9,15,100,101]. For the first role, there is in vitro evidence of LLO causing transient damage to the cell membranes in HBMEC, where this temporary barrier disruption could help *Lm* in traversal of the BBB in a paracellular manner [102]. Although this seemingly contradicts previous statements about the optimal pH for LLO activity being 5.5, it has to be noted that LLO is not inactive at pH higher than 5.5, but rather less active [91,92,100]. However, it was also reported that LLO activity can initiate the uptake of *Lm* directly via a signaling cascade, either alone or in cooperation with Inl family members, suggesting a role in transcellular transport of bacteria across the BBB as well [1,36,100,101]. Concerning the second role, a recent finding shows that the intranasal infection of newborn mice with *Lm* invariably results in invasion of the brain via the olfactory epithelium and associated nerves but only if the bacteria are expressing functional LLO, without which there is no further illness beyond superficial infection of mucosa causing a localized rhinitis [15]. Bacteria with deleted InlA, InlB or both were acting in the same manner as the wild type bacteria as long as LLO was expressed, implying it as vital for the success of listerial spread to the CNS by this specific neurogenous route [15].

The basic mode of action of LLO was outlined earlier, but the generation of a pore in the host cell membrane caused by it has several consequences. One of them is the influx of Ca^2+^ into the cell, which can result in both the destabilization of the membrane and in the initiation of a proposed signaling cascade that promotes bacterial internalization [100,101,103]. Specifically, increased concentration of intracellular Ca^2+^ leads to the recruitment of conventional protein kinase C (cPKC) from the cytosol to the cellular membrane, as well as its activation [103]. Phosphorylated cPKC then activates the small Rho GTPase Rac1—mentioned earlier as being activated downstream of Met during InlB-mediated entry into the cells—which results in the recruitment of the Arp2/3 complex and subsequent pre-internalization actin remodeling [42,103]. Conveniently, the increased concentration of intracellular Ca^2+^ also triggers the membrane resealing pathway, preventing excess damage to the cellular membrane [103].

#### 2.2.4. Proteins Encoded by Listerial Pathogenicity Island 4 (LIPI-4)

As mentioned earlier, most of the research on *Lm* is performed with established strains, and studies aimed at identification of novel VF are not an exception. The most commonly used reference strains (EGD, EGDe, LO28 and 10403S), however, belong to clonal complexes which are rarely isolated from patients (CC7 and CC9), which might lead to VF not present in them remaining unidentified [2]. To address this issue, Maury and colleagues utilized a humanized mouse model of *Lm* infection to determine relative virulence of reference strains (EGDe and 10403S) in comparison to clinically relevant strains (belonging to CC1, CC4 and CC6) and strains frequently detected in food samples (belonging to CC9 and CC121) [2]. The clinically relevant strains were much more aggressive than the others, initiating advanced cases of listeriosis characterized by body weight loss and systemic spread to the liver and the brain [2]. These results were in accord with epidemiological data, which indicated that the strains belonging to CC1, CC4 and CC6 were significantly more represented in cases of advanced listeriosis (including invasion of the brain and the fetus) as well as being encountered in immunocompetent people much more often than strains belonging to other CC [2]. Genome comparison of strains shown to be hypervirulent in this model with those which were hypo-virulent unveiled multiple genes found in the former but not in the latter, one of which was a cluster consisting of six genes associated with CC4, which was later renamed as *Listeria* pathogenicity island 4 (LIPI-4) [2].

Not much is known about the role of LIPI-4 in virulence. It was annotated as cellobiose-family phosphotransferase system (PTS), linking it to the uptake of cellobiose [104]. To assess the impact of LIPI-4 on the efficiency of brain invasion, a LIPI-4 deletion mutant was generated in LM09-00558 (referent CC4 strain) and tested alongside wild type LM09-00558 and EGDe. While the deletion had no impact on bacterial loads in the blood and the liver, the number of bacteria in the brain as well as the placenta was significantly lower compared to the wild type, in the range of neuro-invasively weak strain EGDe. Complementation of the mutant with wild type LIPI-4 reverted the effects of deletion [2].

Since *Lm* frequently finds itself in surroundings with very varying levels and types of available sugars, components of its carbohydrate metabolizing machinery could play a crucial part in keeping the bacterium alive and giving it a competitive advantage in comparison to bacterial strains which do not express these components. Still, more in-depth research into LIPI-4 is required before any conclusions can be made [2,104].

## 3. Conclusions

Most of the research concerning bacterial neuro-invasiveness focused on the hematogenous routes of brain invasion. There are a number of bacterial species that are able to invade the CNS, but the majority of CNS infections are caused by only a few of them. Neuro-invasive strains of *E. coli* (*Ec*) and Group B *Streptococcus* (GBS) are prevalent in cases of neonatal bacterial meningitis; *Haemophilus influenzae* (*Hi*), *Neisseria meningitidis* (*Nm*) and *Streptococcus pneumoniae* (*Sp*) are associated with bacterial meningitis in children and adults; and *Lm*-induced meningitis is found mainly in neonates, elderly and otherwise immunocompromised people [105]. Although the entry points of these bacteria into the body vary (respiratory tract for *Hi*, *Nm* and *Sp* and gastrointestinal tract for *Ec*, GBS and *Lm*), they share a similar general strategy to reach the CNS, necessitated by the constrains put on them by the structure of the vascular system and the brain: 1. adhesion to a mucosal surface (respiratory or gastrointestinal); 2. crossing of the epithelial barrier into the blood; 3. persistence in the bloodstream; 4. adhesion to microvascular endothelium/CP epithelium in the brain; 5. crossing across the blood-brain barriers [105,106]. There are differences in the approach of various bacteria at all of these stages based on the specific VF utilized, but the most relevant stages for the entry into the brain are the last two: adhesion to and penetration of the brain barriers. The adhesion step is administered by interaction of capsule components, pili, fimbriae and adhesins on the surface of bacteria (Inl family members in case of *Lm*) with multiple host cell molecules (e.g., receptors and proteoglycans), with some bacteria colonizing the surface of the cell layer before the invasion (*Nm*) [105,106]. The invasion step itself is the more diverse of the two—most neuro-invasive bacterial species can assumingly utilize both the paracellular and the transcellular pathway, but some seem to use only one of them (paracellular for GBS and likely transcellular for *Lm*). Paracellular entry can be achieved either by “unlocking” of the tight junctions (*Nm*) or by disruption of the barriers through killing of the cells in the layer with toxins (Hcp1 in *Ec*) or cytolysins (β-hemolysin in GBS and pneumolysin in *Sp*) —interestingly, LLO, the signature cytolysin of *Lm*, does not seem to have this role [91,92,93,94,105,106]. Transcellular entry can be direct or indirect. While the indirect entry is based on exploitation of parasitized phagocytes, the direct entry is dependent on the induced “zipper-type” endocytosis of the bacteria through interaction with the target cell mediated by binding of bacterial surface proteins to specific cellular receptors (e.g., IbeA in *Ec*, InlA in *Lm*) [105,106].

*Lm* is a versatile pathogen, able to thrive within its human and animal hosts. Unsurprisingly, it has an arsenal of VF at its disposal that provides it with increased survivability and access to otherwise unreachable niches within the body. Many of these VF overlap in function or complement each other, assuring that the bacterium has backups and specialized tools at hand for various situations. As can be concluded from observing the example of listerial CNS entry, the adaptability of its toolbox leads to different possible strategies and solutions in overcoming the problems it encounters—in this case, the defenses preventing the spreading into the brain. Identification of the main route of entry has been an open question for decades, but the answer to it might be that there is no single main route. Two main proposed routes of entry—hematogenous and neurogenous, including all the variants of both—are not mutually exclusive, and it is imaginable that the one that will be favored in any particular case is selected opportunistically, based on the ease of its application at the time of infection.

The overview of listerial virulence is still fragmented. Raising awareness of the differences between clinically and environmentally associated CC led to the discovery of novel VF (not present in reference strains most commonly used in research) through the use of comparative genetics, which seems to be a promising approach for gaining more insight into the *Lm* virulence machinery [2]. Something that has to be considered as well is the relatively low amount of knowledge currently available on the majority of already identified listerial VF: until their role and mechanisms of action have been properly described, it will be difficult to explore all the paths *Lm* undertakes on its way to the brain. When it comes to understanding and tracking of the bacterial entry into and the spread within the brain, there are several routes along which future research could be directed. Firstly, the development of new animal CNS listeriosis models should continue, leading to in vivo study data more compatible with data obtained from clinical cases in humans and ruminants. Secondly, more precise live imaging and tracking techniques are necessary for researchers to be able to identify, locate and follow the spread of *Lm* within the brain as the standard MRI might not be best suited for this purpose. Thirdly, there always remains the need for novel in vitro solutions, with the application of organotypic brain slices for infection experiments being a good example [107].

All figures in this paper were created with BioRender (https://biorender.com/).

## Figures and Tables

**Figure 1 toxins-12-00297-f001:**
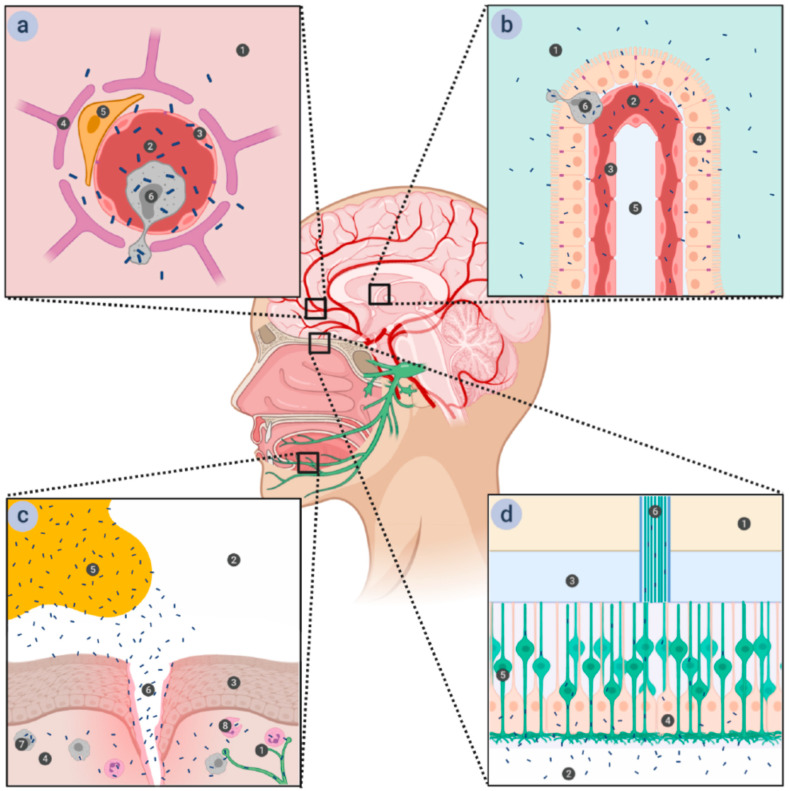
Various proposed routes for listerial entry into the brain. (**a**) Entry via the blood-brain barrier (BBB). The BBB is comprised of the microvascular endothelial cells connected by tight junctions and supported by astrocytes and pericytes. Passage through the microvascular endothelium proceeds in a transcellular manner, with the bacteria either inducing endocytosis into the cells by interacting with the receptors on the cellular surface or being carried across it in infected monocytes. 1—brain tissue; 2—microvascular capillary lumen; 3—microvascular endothelium; 4—astrocyte; 5—pericyte; 6—macrophage. (**b**) Entry via the blood-cerebrospinal fluid barrier (BCSFB). The BCSFB is comprised of the choroid plexus epithelial cells connected by tight junctions—the endothelium of the choroid plexus is fenestrated and presents no obstacles to the bacteria. Passage through the choroid plexus epithelium is transcellular, either direct—by induction of endocytosis into the cells—or indirect—within infected monocytes, similar to passage through the BBB. 1—cerebrospinal fluid (CSF); 2—choroid plexus capillary lumen; 3—choroid plexus endothelium; 4—choroid plexus epithelium; 5—connective tissue; 6—macrophage. (**c**) Entry via the trigeminal nerve. Damage to the stratified squamous epithelium that forms the top layer of oral mucosa opens a path for bacteria from contaminated food to enter the submucosa, where they can be phagocytized by either resident macrophages or recruited phagocytes. *Lm* is able to survive within these cells and to spread further to the nerve endings of the trigeminal nerve, along which it can travel to the brain stem. 1—trigeminal nerve ending; 2—oral cavity; 3—stratified squamous epithelium of the oral mucosa; 4—submucosa; 5—*Lm*-contaminated food; 6—wound in the oral mucosa; 7—resident macrophage; 8—recruited phagocyte. (**d**) Entry via the olfactory epithelium. The ciliated endings of olfactory sensory neurons are located at the surface of olfactory epithelium, supported by epithelial cells and covered in mucus. The axons of olfactory sensory neurons form bundles that pass through the ethmoid bone of the skull and end in the olfactory bulb. *Lm* can access these neurons when liquids contaminated with *Lm* (such as vaginal secretions ingested during childbirth) are ingested into the nasal cavity, and traveling along them provides a direct route to the brain. 1—olfactory bulb; 2—nasal cavity; 3—ethmoid bone; 4—supporting epithelium; 5—olfactory neuron; 6—olfactory nerve fiber.

**Figure 2 toxins-12-00297-f002:**
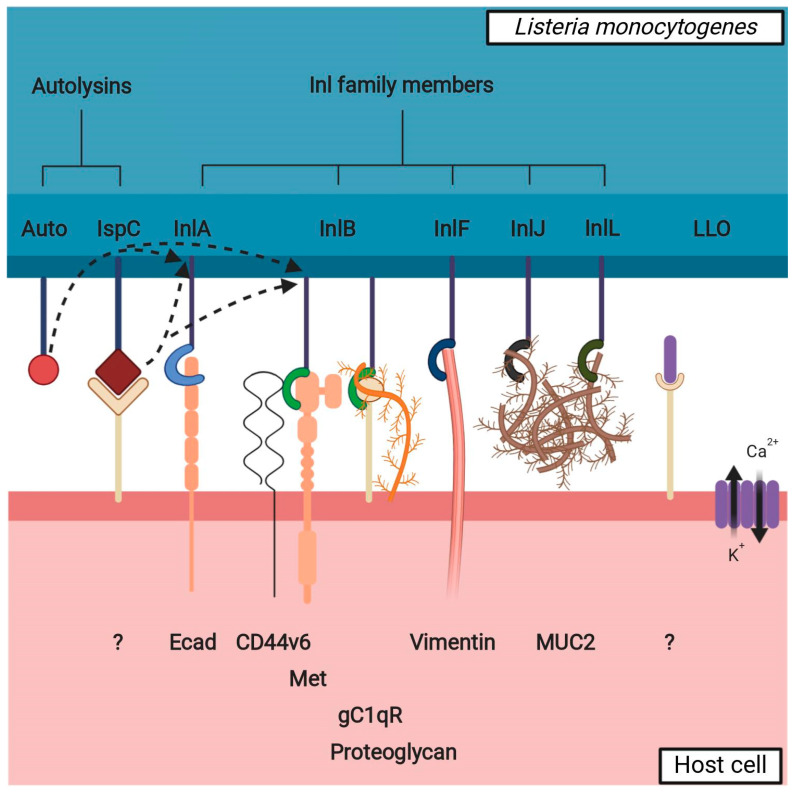
Interaction between listerial VF potentially involved in brain invasion and a non-phagocytic host cell. Inl family members bind to cellular surface receptors to assist *Lm* in adhesion to and/or invasion into the cell. The interaction partners of presented Inl family members are readily expressed in the brain with the exception of MUC2, which is mainly expressed in the intestinal mucosa—it is possible that InlJ and InlL have other interaction partners in the brain. Autolysins (Auto, IspC) participate in the remodeling of the peptidoglycan layer, affecting the presence of the Inl family members on the bacterial surface (shown by dotted arrows); IspC also has a role in adhesion to the cellular surface. LLO oligomerizes to form pores on the cell membrane and thus destabilizes it, with a twofold effect: influx of Ca^2+^: a) causes the disruption of tight junctions between the cells (paracellular path) and b) begins a signaling cascade resulting in pre-endocytosis actin remodeling (transcellular path); LLO also possibly interacts with a cellular surface receptor to initiate a downstream cellular signal. Unknown or hypothetical interaction partners on cellular surface are labeled with a question mark.

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
