# Peer review of "Potential Roles and Functions of Listerial Virulence Factors during Brain Entry"

_toxins, 2020, doi:10.3390/toxins12050297_

Round 1
Reviewer 1 Report
Comments to the Authors
This manuscript provides an overview of the current knowledge on how the bacterial pathogen Listeria monocytogenes enters into the brain, which is a key - yet understudied - step in neurolisteriosis. The authors also discuss the potential role of several virulence factors in this process. So far, no comprehensive review is available on this topic and the article is of great interest for the scientific community. The graphical illustrations are well conceived. However, some information is missing and the review would benefit of editing to eliminate writing style errors (throughout the text). Significant revision is therefore needed to improve it.
Specific comments
- Title. The title of the review is not entirely accurate since the role of listerial virulence factors in brain entry most often remains characterized at the hypothetical stage.
- Abstract
- line 10. The majority of studies were not only focused on the gut barrier but also on the maternal-fetal barrier.
- line 11. Change "Here we'll assemble the current knowledge... " into " Here, we review current knowledge...".
- Introduction
- line 24. When compared to other bacterial species, L. monocytogenes is not a genetically VERY diverse species. Remove "very".
Moreover, in addition to the old classification into lineages, L. monocytogenes strains are now classified into clonal complexes. Please add some discussion about it (which is in reference 2, Maury et al. 2016).
- line 25-26. In immunocompetent people, T cell–mediated immunity is highly efficient in clearing L. monocytogenes infection. In addition, there is an asymptomatic carriage, which is probably much more important than gastroenteritis. This information should be added (see discussion in the review Bierne et al. Front Cell Infect Microbiol. 2018 May 15;8:136. PMID: 29868493).
- line 33. It should be noted that in the liver, Listeria is mainly eliminated by resident macrophages, especially Kupffer cells. Only a fraction of bacteria survive in hepatocytes.
- line 36. Invasion of the placental-fetal unit by L. monocytogenes should be mentioned here.
- line 73. Some clarification is needed about the differences in the manifestation of neurolisteriosis in humans and animals. What is the part of meningitis, meningo-encephalitis, rhombencephalitis and brain abscesses in humans and ruminants?
Also, please indicate when conclusions derive from experimental studies (for instance, in mice). In line with this, the authors should add and discuss the following reference:
Antal et al. Evidence for intraaxonal spread of Listeria monocytogenes from the periphery to the central nervous system. Brain Pathol. 2001 Oct;11(4):432-8. PMID: 11556688).
- Virulence factors
- line 118. There are 25 members of the internalin family in strain EGDe, but there are other members in other strains, particularly in clinical strains (see reference 22, Bierne et al. 2007). Please, update this information. The authors may cite the following recent example: Harter et al. The Novel Internalins InlP1 and InlP4 and the Internalin-Like Protein InlP3 Enhance the Pathogenicity of Listeria monocytogenes. Front Microbiol. 2019 Jul 23;10:1644. PMID:31396177).
- line 119-120: Internalins exhibit a Sec-dependent N-terminal signal peptide and a LRR domain.
- line 122: The second class of internalins comprise two proteins, InlB and Lmo0549. Both display a C-terminal domain that directs non-covalent association to the cell surface, but Lmo0549 is not reported to bind lipoteichoic acid. It could be to another cell wall component.
- line 122: The majority of Inl proteins is "relatively poorly described and characterized"... but InlA and InlB have been extensively characterized, host partners and functions have been identified for InlC, InlF, InlK and InlP, and there are some data on the potential function of InlH, InlJ and InlL.
- line 255: the fact that the LRRs of InlJ contain cysteine does not indicate that InlJ interacts with different type of host proteins. It only points to an original structure.
(Bublitz M, et al. Crystal structure and standardized geometric analysis of InlJ, a listerial virulence factor and leucine-rich repeat protein with a novel cysteine ladder.J Mol Biol. 2008 Apr 18;378(1):87-96).
- line 266. InlJ contains a MucBP domain and purified InlJ has been shown to bind to mucin MUC2
Lindén et al. 2008. Listeria monocytogenes internalins bind to the human intestinal mucin MUC2. Arch Microbiol. 2008 Jul;190(1):101-4. doi: 10.1007/s00203-008-0358-6. Epub 2008 Mar 8.
- line 271. Data on InlL (Lmo2026, and formerly ORF626) are missing. By screening a bank of signature-tagged transposon mutants in mouse model, Autret et al. suggested that ORF626 could affect listerial multiplication in the brain (Autret et al., 2001). However, research is needed to confirm a possible role of this internalin in the crossing of the blood–brain barrier.
Autret et al. 2001. Identification of new genes involved in the virulence of Listeria monocytogenes by signature-tagged transposon mutagenesis. Infect Immun. 2001. Apr;69(4):2054-65.
Like InlJ, InlL is not present in non-pathogenic Listeria species. It has a MucBP domain and binds to MUC2.
Popowska et al. InlL from Listeria monocytogenes Is Involved in Biofilm Formation and Adhesion to Mucin. Front Microbiol. 2017 Apr 20;8:660.
Of note, together with InlJ and InlL, seven internalins contain MucBP in L. monocytogenes, namely InlI (Lmo0333), Lmo0171, Lmo0327, Lmo0732 and Lmo2396. While interactions with mucins are require for many enteric pathogens to cause infection, the contribution of these internalins to the physiopathology of L. monocytogenes still requires in-depth investigations, in line with the presence of mucins in the gastro-intestinal tract, but also in other organs. Mucin could play a role in bacterial adhesion to epithelial cells in the brain barriers.
- line 358. Regarding other virulence factors, the authors need to discuss the input of comparative genomics to the topic. Specifically, they should discuss the important finding of Maury et al. 2016 (reference 2). L. monocytogenes clones epidemiologically associated with human central nervous system (CNS) and maternal neonatal listeriosis (namely CC1, CC4 and CC6) happen to be most prevalent in patients without immunosuppressive comorbidities, and, consistent with this observation, are hypervirulent in a humanized mouse model of listeriosis. In contrast, the clones most prevalent in food and least prevalent in patients (namely CC9 and CC121) are hypovirulent. Comparative genomics between hypervirulent and hypovirulent clones led to the identification of multiple new putative virulence factors. One gene clusters, which encodes a putative PTS, has been shown to be involved in the enhanced neural and placental tropism of CC4, by a mechanism that remains to be identified (Maury et al., 2016).
- Conclusions Page 9.
The authors may propose some ideas for future research directions: for instance on the need of animal models, imagery techniques or tissue microbiology (For instance, brain organotypic brain-slices. see Guldimann C, et al. 2015. Increased spread and replication efficiency of Listeria monocytogenes in organotypic brain-slices is related to multilocus variable number of tandem repeat analysis (MLVA) complex. BMC Microbiol. 2015 Jul 3;15:134).
Additionally, a brief discussion of Listeria entry into the brain in comparison with other pathogens would be helpful.
see: Le Guennec et al. Strategies used by bacterial pathogens to cross the blood-brain barrier.Cell Microbiol. 2020 Jan;22(1):e13132. PMID:31658405
Herold et al. Virulence Factors of Meningitis-Causing Bacteria: Enabling Brain Entry across the Blood-Brain Barrier. Int J Mol Sci. 2019 Oct 29;20(21). pii: E5393. PMID:31671896
- Figure 2.
In the title of this figure, the authors should add "potentially", as follows: "Interaction between listerial VF potentially involved in brain invasion". Indeed, a clear demonstration of the role of these factors in the brain is still missing. In addition, the figure should include other host ligands for InlB (Gc1qR and proteoglycans) and InlL (beside InlJ). Both InlJ and InlL potentially interact with host mucins. This could be added with a question mark.
Reviewer 2 Report
In this review, the authors provide a comprehensive and thoroughly referenced compilation of the current knowledge regarding routes and mechanisms of L. monocytogenes infection of the central nervous system, in particular the brain, as well as of the bacterial virulence factors that have been shown to play roles in these processes.
I have only minor comments/corrections for the authors:
- Line 5: It is the disease (listeriosis) that is rare, not the bacterium,
- Line 11: It is common etiquette in scientific writing not to use word contractions such as “we’ll”. Please, replace with “we will” or use the present tense “we assemble”.
- Lines 24 and 360: “very” is unnecessary.
- Line 118: Italicize “Listeria”.
- Lines 137 and 369: “it is”
- Lines 221 and 224: “did not”
- Lines 267-268: Italicize “Listeria innocua”.
- Lines 105, 133, 190, 257, 313, 372: Replace spelled-out “virulence factor”/”virulence factors” by “VF”.
- Line 326: Italicize “hly”.
- Line 328: “does not”
- Figure 1 legend: Please spell out all abbreviations (BBB, BCSFB, CSF) in their first appearance. The reader may go to the figure legend before knowing what these abbreviations mean in the body text.
Round 2
Reviewer 1 Report
The authors have satisfactorily responded to my concerns.
Please note that although I accept the paper in its present form, some typos errors remain. And abbreviations (BBB and BCSFB line 85 page 3) should be defined.
Regarding Figure 2, it would be helpful to find in the legend, a comment on the fact that MUC2 is mainly expressed in the intestinal mucosa, and that other mucins and/or host ligands could be targeted by InlJ and InlL in the brain.
Author Response
Please note that although I accept the paper in its present form, some typos errors remain. And abbreviations (BBB and BCSFB line 85 page 3) should be defined.
We corrected the typographical errors we found.
Line 88-89: The abbreviations BBB and BCSFB are now defined in the text, additionally to the legend of Figure 1.
Regarding Figure 2, it would be helpful to find in the legend, a comment on the fact that MUC2 is mainly expressed in the intestinal mucosa, and that other mucins and/or host ligands could be targeted by InlJ and InlL in the brain.
Legend of Figure 2: We have now added the mention of other possible interaction partners for InlJ and InlL, due to the primarily intestinal localization of MUC2.